# THE UNSEEN BIAS: HOW NORM DISCREPANCY IN PRE-NORM MLLMS LEADS TO VISUAL INFORMATION LOSS

**Bozhou Li**[1,6,*] **Xinda Xue**[1], **Sihan Yang**[1,5], **Yang Shi**[1],
**Xinlong Chen**[4], **Yushuo Guan**[6], **Yuanxing Zhang**[6], **Wentao Zhang**[1,2,3,†]
[1]Peking University
[2]Zhongguancun Academy
[3]Beijing Key Laboratory of Data Intelligence and Security (Peking University)
[4]School of Artificial Intelligence, University of Chinese Academy of Sciences
[5]Xi'an Jiaotong University
[6]Kling Team, Kuaishou Technology

## ABSTRACT

Multimodal Large Language Models (MLLMs), which couple pre-trained vision encoders and language models, have shown remarkable capabilities. However, their reliance on the ubiquitous Pre-Norm architecture introduces a subtle yet critical flaw: a severe norm disparity between the high-norm visual tokens and the low-norm text tokens. In this work, we present a formal theoretical analysis demonstrating that this imbalance is not a static issue. Instead, it induces an "asymmetric update dynamic," where high-norm visual tokens exhibit a "representational inertia," causing them to transform semantically much slower than their textual counterparts. This fundamentally impairs effective cross-modal feature fusion. Our empirical validation across a range of mainstream MLLMs confirms that this theoretical dynamic—the persistence of norm disparity and the resulting asymmetric update rates—is a prevalent phenomenon. Based on this insight, we propose a remarkably simple yet effective solution: inserting a single, carefully initialized LayerNorm layer after the visual projector to enforce norm alignment. Experiments conducted on the LLaVA-1.5 architecture show that this intervention yields significant performance gains not only on a wide suite of multimodal benchmarks but also, notably, on text-only evaluations such as MMLU, suggesting that resolving the architectural imbalance leads to a more holistically capable model.

## 1 INTRODUCTION

In recent years, Multimodal Large Language Models (MLLMs) have achieved significant progress, demonstrating robust performance across a wide range of cross-modal tasks (Comanici et al., 2025; Hurst et al., 2024; Wu et al., 2024; Bai et al., 2025). A prevailing architectural paradigm involves augmenting a pre-trained Large Language Model (LLM) with visual capabilities by coupling it with a pre-trained Vision Encoder (VE). The VE, typically a Vision Transformer (ViT) (Dosovitskiy et al., 2020), first partitions an image into a sequence of patches and encodes them into a series of feature vectors, or "visual tokens." To bridge the modality gap, a lightweight adapter module is then introduced. This module's core function is to act as a translator, projecting these visual tokens into the LLM's word embedding space, thereby making visual information comprehensible to a model originally designed for text (Zhang et al., 2024).

Despite their powerful general-purpose capabilities, emerging research has revealed inherent limitations in MLLMs. For instance, many models struggle with the perception of fine-grained visual details (Rahmanzadehgervi et al., 2024). Furthermore, within their self-attention mechanisms—the core component for weighing the importance of different inputs—visual tokens often receive less

---

[*]Email: libozhou@pku.edu.cn
[†]Corresponding author:Wentao Zhang

focus than their textual counterparts (Chen et al., 2024a). To address these challenges, we identify a more fundamental problem rooted in the now-ubiquitous Pre-Norm Xiong et al. (2020) architectural design. In this paradigm, normalization is applied before the main computational block ($F$), with the residual update defined as:

$$\mathbf{h}^{(l+1)} = \mathbf{h}^{(l)} + F(\text{Norm}(\mathbf{h}^{(l)})) \tag{1}$$

This architecture is widely adopted because it is easier to train. By leaving the residual path $\mathbf{h}^{(l)}$ unaltered, it creates an identity-like connection that ensures smooth gradient flow, preventing vanishing gradients in deep networks. However, this design has a critical side effect: since the output of the residual sum is never re-normalized, the variance—and consequently, the $L_2$ norm—of the hidden states tends to accumulate and grow with network depth (Kim et al., 2025). As is shown in Figure 1b, it creates a particularly acute imbalance in MLLMs where high-norm visual tokens and lower-norm text tokens are processed together within a shared Pre-Norm LLM backbone—as the visual tokens themselves are generated by a deep, Pre-Norm ViT.

Our formal theoretical analysis reveals a critical dynamic: a fundamental asymmetry in the evolutionary pace of visual and textual representations through the LLM's layers. We demonstrate that for high-norm visual tokens, the Pre-Norm update mechanism induces a high "representational inertia", causing them to undergo a much slower semantic transformation. In contrast, lower-norm textual tokens adapt their representations more readily, leading to a mismatched rate of convergence towards a unified multimodal space. Notably, this dynamic divergence arises not from an intrinsic property of visual versus textual information, but from an architectural artifact: the interplay between the Pre-Norm design and the prevailing MLLM paradigm.

Bridging theory and practice, we first confirmed that these norm disparities and asymmetric update rates are prevalent across mainstream open-source VL models. Based on this validation, we propose a targeted intervention: inserting a normalization layer to enforce strict norm alignment. However, practical implementation reveals a critical optimization bottleneck. Since text embeddings in modern LLMs (e.g., Qwen2.5) exhibit extremely low magnitudes, aligning visual tokens to this target requires initializing the normalization gain to a minute value. This naively triggers a *vanishing gradient problem*, detaching the vision encoder from supervision. To resolve this, we introduce a Global Weight Compensation (GWC) mechanism. This technique decouples the forward norm compression from the backward gradient magnitude, ensuring effective learning even under extreme alignment constraints.

In this work, our key contributions are threefold:

- **Theoretical Identification of Asymmetric Dynamics.** We are the first to identify and theoretically formalize the issue of cross-modal norm disparity in Pre-Norm MLLMs. Our analysis reveals an "asymmetric update dynamic" where high-norm visual tokens exhibit "representational inertia," leading to a slower semantic evolution compared to text tokens.

- **Extensive Empirical Validation.** We provide extensive empirical validation across a suite of mainstream open-source MLLMs, demonstrating that the predicted norm disparities and asymmetric update rates exist, confirming our theoretical model in practice.

- **A Simple and Robust Solution.** We propose Gradient-Aware Norm Alignment, incorporating a novel WC mechanism. This approach resolves the optimization dilemma caused by extreme norm compression. Our experiments show that this method yields significant performance gains not only on multimodal tasks but also, notably, on text-only benchmarks, indicating a more holistic improvement to the model's capabilities.

## 2 PRELIMINARIES

Our analysis is grounded in the core components of modern Transformers: self-attention, normalization, and the choice of residual architecture.

**Self-Attention and Normalization Layers.** The self-attention mechanism projects input sequences into queries ($\boldsymbol{Q}$), keys ($\boldsymbol{K}$), and values ($\boldsymbol{V}$) to compute unnormalized dot-product scores

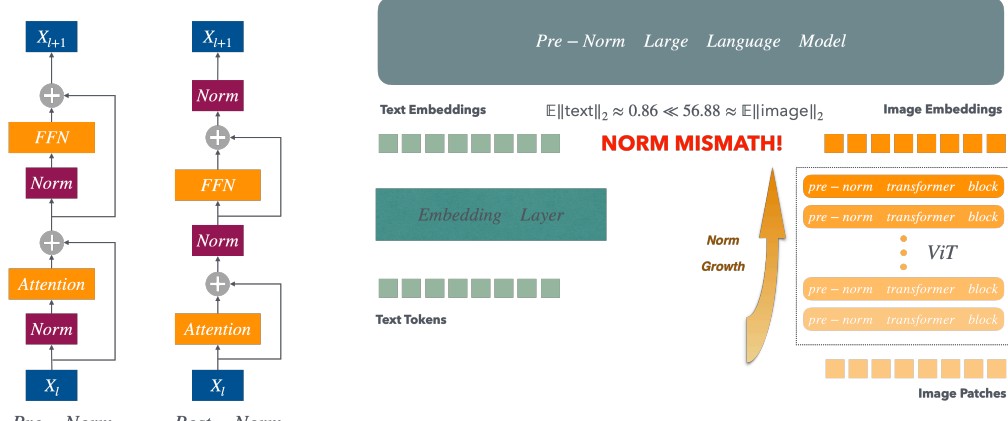

(a) Pre-Norm vs Post-Norm        (b) The Norm Mismatch Problem Induced by MLLM Architectures

$QK^T$. To stabilize training and manage activation scales, normalization is applied. **LayerNorm** (Ba et al., 2016) normalizes features per token, where $g$ and $\beta$ are learnable gain and bias parameters:

$$\text{LayerNorm}(\boldsymbol{x}) = \frac{\boldsymbol{x} - \mathbb{E}[\boldsymbol{x}]}{\sqrt{\text{Var}[\boldsymbol{x}] + \epsilon}} \odot \boldsymbol{g} + \boldsymbol{\beta}$$

**RMSNorm** (Zhang & Sennrich, 2019) is a computationally efficient variant that omits mean-centering and is widely adopted in modern LLMs:

$$\text{RMSNorm}(\boldsymbol{x}) = \frac{\boldsymbol{x}}{\sqrt{\frac{1}{D}\|\boldsymbol{x}\|_2^2 + \epsilon}} \odot \boldsymbol{g}$$

**Pre-Norm vs. Post-Norm Residuals.** A Transformer block refines representations via an additive update: $\boldsymbol{h}^{(l+1)} = \boldsymbol{h}^{(l)} + \Delta\boldsymbol{h}^{(l)}$. The placement of normalization dictates the network's training dynamics:

- **Post-Norm**: Normalizes after the residual sum, preserving strong representational fidelity but often impeding gradient flow in deep networks:

$$\boldsymbol{h}^{(l+1)} = \text{Norm}\big(\boldsymbol{h}^{(l)} + \text{Sublayer}(\boldsymbol{h}^{(l)})\big)$$

- **Pre-Norm**: Normalizes within the residual branch, leaving the skip connection unaltered:

$$\Delta\boldsymbol{h}^{(l)} = \text{Sublayer}(\text{Norm}(\boldsymbol{h}^{(l)}))$$
$$\boldsymbol{h}^{(l+1)} = \boldsymbol{h}^{(l)} + \Delta\boldsymbol{h}^{(l)}$$

This uninterrupted connection guarantees stable gradients. However, because the residual output $\boldsymbol{h}^{(l+1)}$ bypasses normalization, the variance—and consequently the $L_2$ norm—monotonically accumulates with depth (Kim et al., 2025).

Building on these definitions, we formalize our central argument: the Pre-Norm design inherently creates a destructive dynamic imbalance in MLLMs. By injecting features from a deep, pre-trained vision encoder (where norms have already accumulated) into the relatively low-norm embedding space of an LLM, a severe initial $L_2$ norm disparity is established exactly at the cross-modal interface.

## 3   THEORETICAL ANALYSIS OF NORM-INDUCED DECOUPLING EFFECT

In this section, we present a theoretical proof that this initial norm imbalance is not a static issue but rather the catalyst for an accelerated geometric divergence between the two modalities, ultimately suppressing the cross-modal attention signal. The full mathematical derivation is provided in the Appendix.

### 3.1 ANALYTICAL FRAMEWORK AND ASSUMPTIONS

Our proof is predicated on a set of simplifying assumptions that capture the core dynamics of the Pre-Norm architecture:

- **Modality Norm Imbalance**: We analyze two cases: the **imbalanced case** ($k = \frac{\|\boldsymbol{h}_{\text{vis}}\|_2}{\|\boldsymbol{h}_{\text{txt}}\|_2} > 1$) and the ideal **balanced case** ($k = 1$).

- **Uniform Update Magnitude**: Due to the Pre-Norm design, the magnitude of the update vector, $\|\Delta \boldsymbol{h}^{(l)}\|_2$, is decoupled from the input norm $\|\boldsymbol{h}^{(l)}\|_2$. We denote this uniform magnitude as $C^{(l)}$ for a given layer.

- **Consistent Update Geometry**: We assume the update vector $\Delta \boldsymbol{h}$ forms a consistent expected angle, $\phi$, with the hidden state $\boldsymbol{h}$ for all tokens within a given layer.

- **Random Rotational Direction**: We assume the direction of the rotational component of the update is drawn from a symmetric distribution over the relevant subspace.

### 3.2 ASYMMETRIC ANGULAR VELOCITY AND GEOMETRIC DIVERGENCE

To quantify the rate of directional change, we introduce the concept of **effective angular velocity**. The update vector $\Delta \boldsymbol{h}$ can be decomposed into a component parallel to the hidden state $\boldsymbol{h}$ (which only scales its length) and a component orthogonal to it (which causes rotation). The effective angular velocity, measured by the angle of pure rotation $\theta_{\text{eff}}$, is driven solely by this orthogonal component. As derived in Appendix B, its tangent is given by:

$$\tan(\theta_{\text{eff}}) = \frac{C^{(l)} \sin(\phi)}{\|\boldsymbol{h}\|_2 + C^{(l)} \cos(\phi)} \tag{2}$$

A direct and critical consequence of our framework is that this angular velocity becomes asymmetric in the imbalanced case. Because a uniform update magnitude $C^{(l)}$ is applied to hidden states of different norms, the high-norm vision tokens exhibit a lower effective angular velocity than the low-norm text tokens. Formally, for $\|\boldsymbol{h}_{\text{vis}}\|_2 > \|\boldsymbol{h}_{\text{txt}}\|_2$, it follows that:

$$\tan(\theta_{\text{eff, vis}}) < \tan(\theta_{\text{eff, txt}}) \tag{3}$$

This disparity imparts a higher "representational inertia" to visual tokens. In Appendix B, we rigorously prove that this asymmetry leads to an accelerated geometric divergence between the representations of the two modalities, which in turn weakens the underlying similarity signal available to the attention mechanism.

### 3.3 SIGNAL DEGRADATION AND ATTENTION COLLAPSE

The geometric divergence derived above has a specific and destructive consequence for the attention mechanism: it prevents the semantic alignment of related tokens. In a functioning multimodal model, a text query $\mathbf{q}$ seeks a semantically relevant visual key $\mathbf{k}^+$. Specifically, this retrieval process involves evolving the hidden states into a configuration where their projections are geometrically aligned in the shared metric space, allowing their dot product to be maximized.

However, the "representational inertia" creates a mechanical barrier to this alignment. While the low-norm text token rapidly updates its orientation to query the image, the high-norm visual token updates too slowly to reciprocate. This results in a persistent *angular misalignment* for semantically related pairs. Mathematically, the dot product signal for a relevant pair, $S_{\text{rel}} \propto \mathbf{q} \cdot \mathbf{k}^+$, is fundamentally capped by this geometric lag.

In contrast, for irrelevant pairs (noise), the vectors remain approximately orthogonal in high-dimensional space regardless of the inertia. The critical failure mode is thus a collapse in the Signal-to-Noise Ratio (SNR). Because the score of the relevant visual token $S_{\text{rel}}$ is suppressed while the background noise levels remain static, the attention mechanism—governed by the Softmax operation—loses its ability to sharply distinguish the target visual region. Formally, we prove in Appendix B.4 that for the expected attention scores of semantically relevant pairs:

$$\mathbb{E}[S_{\text{rel}}^{(\text{imb})}] < \mathbb{E}[S_{\text{rel}}^{(\text{bal})}] \tag{4}$$

This provides a first-principles explanation for the diffuse and unfocused attention maps observed in baseline models (Appendix G). The norm imbalance denies the model the geometric agility required to establish strong correlations, leading to the effective loss of visual information.

## 4 EMPIRICAL VALIDATION: PROBING THE DYNAMICS OF NORM IMBALANCE

Our theoretical analysis provides a formal, first-principles explanation for how norm imbalance can impair multimodal fusion. However, this framework relies on a set of simplifying assumptions to ensure analytical tractability, while the dynamics of large-scale MLLMs are considerably more complex. Therefore, to bridge the gap between our idealized model and real-world behavior, we conduct a series of empirical investigations. These experiments are designed to probe whether the core consequences predicted by our theory—namely, the persistence of norm imbalance and the resulting asymmetric update dynamics—manifest in state-of-the-art Pre-Norm MLLMs. Our investigation is guided by the following research questions:

- **RQ1: Existence of Initial Norm Disparity.** Do visual and text tokens exhibit a significant norm mismatch at the modality interface?

  To answer this, we benchmarked the $L_2$ norms from both sides of the modality interface. For the visual modality, we measured the output norms of four representative vision encoders—CLIP (Radford et al., 2021), SigLIP (Zhai et al., 2023), SigLIP-v2 (Tschannen et al., 2025), and MoonViT (Team et al., 2025). For the text modality, we established a baseline by computing the average $L_2$ norm of the text embedding layers from prominent LLMs: Qwen2.5 (Bai et al., 2025), Qwen3 (Yang et al., 2025), and Llama3.2 (Grattafiori et al., 2024). The analysis of vision encoders was conducted on a dataset of 1000 samples drawn from the MMBench, POPE, and MM-Star benchmarks, which serves as the foundation for all subsequent experiments in this section. The combined results are presented in Table 1.

Table 1: $L_2$ norms and hidden dimensions at the modality interface: Vision Encoder outputs vs. LLM text embeddings (mean $\pm$ std).

| Modality | Model | Dimension | Average $L_2$ Norm |
|---|---|---|---|
| Visual | CLIP-ViT-large-patch14 | 1024 | $29.30 \pm 17.12$ |
| | SigLIP-SO-400m-patch14-384 | 1152 | $71.78 \pm 13.95$ |
| | SigLIP2-SO-400m-patch14-384 | 1152 | $59.37 \pm 106.08$ |
| | MoonViT-SO-400M | 1152 | $72.17 \pm 7.13$ |
| Text | Qwen2.5-7B-Instruct | 3584 | 0.80 |
| | Qwen3-8B-Instruct | 4096 | 1.38 |
| | Llama3.2-3B-Instruct | 3072 | 1.09 |

As shown in 1, vision encoder output norms are substantially larger than those of text embeddings. This disparity persists because the encoders' contrastive pre-training—even with a final post-norm—is not designed to align with the norm scale of an external LLM's embedding space.

- **RQ2: The Efficacy of the Adapter.** Does the projection adapter harmonize the initial norm disparity before the tokens enter the LLM backbone?

  Within MLLMs, the projector's role is to map visual tokens into the LLM's textual embedding space. A critical question is whether this process also serves to align their norms. To investigate this, we analyzed a suite of prominent models: LLaVA-v1.5 (Li et al., 2024a), Qwen-2.5-VL (Bai et al., 2025), KimiVL (Team et al., 2025), and GLM-4.1V (Hong et al., 2025). For each model, we measured the $L_2$ norm of visual tokens both before and after the projector and compared them to the text token norm. The results are summarized in Table 2.

  The results in Table 2 reveal a clear spectrum of effectiveness across different projector designs. While sophisticated projectors like those in KimiVL and GLM-4.1V demonstrate a significant capability for norm compression, a substantial disparity between visual and text token norms persists in all analyzed models. This varied effectiveness highlights a key finding: simply inserting a normalization layer within the projector is not a guaranteed solution. With the exception of

Table 2: $L_2$ norms of visual tokens (before and after projector, mean ± std) vs. text tokens.

| Model | Visual (Before Proj.) | Visual (After Proj.) | Text (Embedding) |
|---|---|---|---|
| LLaVA-v1.5 | 28.71 ± 16.87 | 39.96 ± 45.58 | 1.08 |
| Qwen-2.5-VL | 3484.24 ± 3882.82 | 56.88 ± 25.73 | 0.86 |
| KimiVL | 137.93 ± 34.43 | 4.78 ± 2.21 | 0.85 |
| GLM-4.1V | 47.44 ± 4.58 | 4.58 ± 2.03 | 0.80 |

LLaVA-v1.5, all other models incorporate internal norm layers, yet their final output norms differ by an order of magnitude.

This leads to a broader discussion on current design practices. We note that these architectural choices and their impact on cross-modal norm alignment are seldom, if ever, addressed in the models' respective technical reports.

Notably, the systemic inflation of text embedding norms observed in Qwen-2.5-VL and Qwen-2-VL (detailed in Appendix C) corroborates that norm discrepancy is a fundamental bottleneck. This phenomenon suggests that the model is forced to inefficiently adjust its static embedding parameters to passively compensate for the massive norm gap at the modality interface.

- **RQ3: Asymmetry in Update Dynamics.** Do visual and textual hidden states exhibit different update rates, as predicted by our theory of asymmetric angular velocity?

  This question serves as the most direct empirical test of our theory's core mechanism. We use the cosine similarity between consecutive layers ($l-1$ and $l$) as a proxy for the rate of representational change, a metric conceptually linked to angular velocity. A higher similarity score implies a smaller angular change and thus a slower update rate. We computed this metric for both modalities across all layers to determine if a systematic divergence in their update rates exists, as shown in Figure 3.

  The results in Figure 3 confirm our theoretical predictions, revealing a consistent divergence in update rates between visual and text tokens across all analyzed models. Notably, the magnitude of this dynamic asymmetry appears to be directly correlated with the initial norm disparity identified in RQ1 and RQ2. Models with a smaller initial norm gap, such as Kimi-VL and GLM-4.5V, exhibit a less pronounced difference in update rates. Conversely, models with a more severe norm imbalance, like LLaVA-1.5 and Qwen-2.5-VL, demonstrate a significantly larger gap in their update dynamics, providing strong correlational evidence for our theory.

- **RQ4: Norm Discrepancy in Models with Visual Embedding Tables.** Does the significant norm disparity persist in architectures utilizing probabilistic visual tokenization, such as Ovis 2.5 (Lu et al., 2025)?

  To verify whether the norm discrepancy and its dynamic consequences persist across different architectural paradigms, we conducted experiments on Ovis-2.5-9B, a model that employs probabilistic visual tokenization. Unlike LLaVA-style models that project continuous features, Ovis utilizes a discrete visual vocabulary derived from a SigLIP2 encoder.

Table 3: $L_2$ norms in Ovis 2.5. Note that while the visual tokens are strictly clustered (low std), their magnitude remains drastically higher than text tokens.

| Modality | Source | Avg $L_2$ Norm |
|---|---|---|
| Visual | SigLIP2 Vocab (Frozen) | **64.00** ± 0.71 |
| Text | Text Vocab (Base LLM) | **1.38** ± 0.32 |

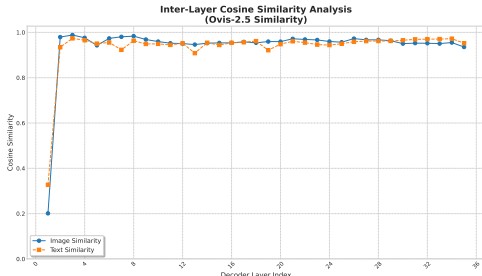

Figure 2: Layer-wise cosine similarity in Ovis 2.5. Despite the massive norm gap (64 vs 1.38), the update rates of visual and text tokens remain remarkably synchronized.

As detailed in Table 3, a substantial static norm discrepancy persists in Ovis 2.5: the visual vocabulary exhibits an average norm of 64.00—likely stemming from its derivation via the SigLIP2 encoder—compared to the text norm of 1.38.

However, Figure 2 indicates that the internal dynamics differ from those observed in continuous projection models. Despite the norm disparity, the update rates (proxied by cosine similarity) for visual and text tokens do not exhibit the marked asymmetry found in LLaVA-style architectures. The curves overlap significantly, suggesting that the discrete tokenization paradigm may inherently mitigate the representational inertia usually associated with high-norm inputs.

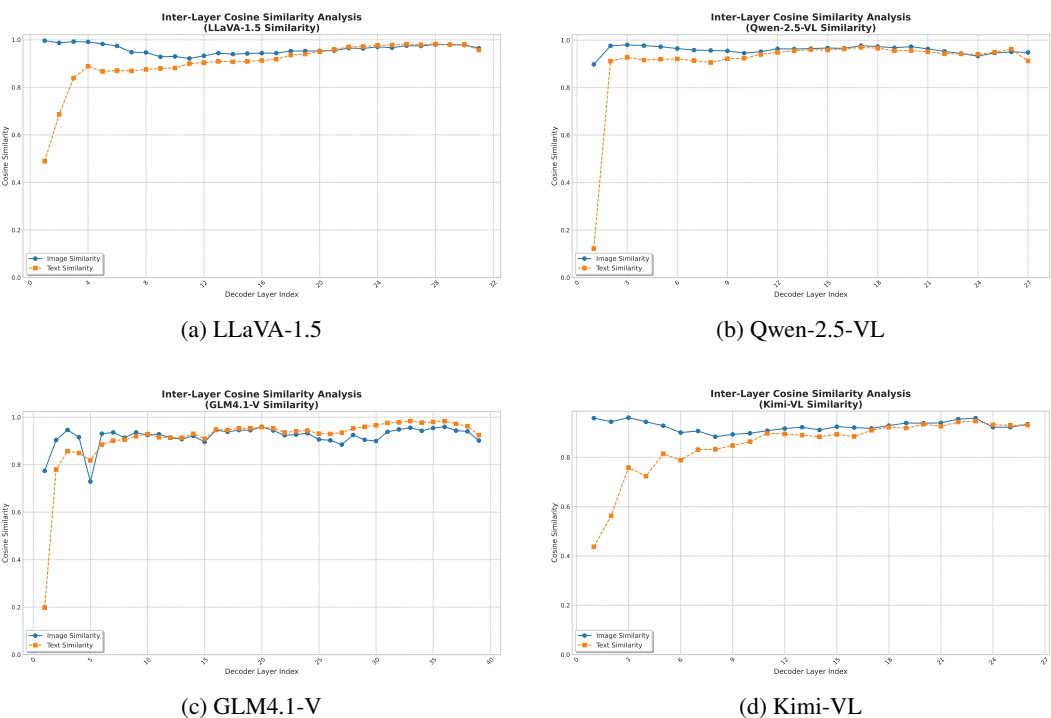

(a) LLaVA-1.5              (b) Qwen-2.5-VL

(c) GLM4.1-V              (d) Kimi-VL

Figure 3: Inter-layer cosine similarity of hidden states for visual vs. text tokens.

## 5 EXPERIMENTS

Our theoretical analysis in Section 3 posited a mechanism whereby norm disparity leads to update asymmetry and, consequently, suppressed visual attention. A critical question remains, however: do these internal dynamics translate into a tangible degradation of the model's downstream capabilities? To investigate this link between internal mechanics and practical performance, we conducted a series of comparative experiments.

### 5.1 METHOD: GRADIENT-AWARE NORM ALIGNMENT

To enforce norm alignment between visual and text tokens, we introduce a straightforward intervention: inserting an additional LayerNorm layer immediately after the visual projector. Crucially, the learnable gain parameter $\mathbf{g}$ of this layer is initialized to explicitly match the average $L_2$ norm of the text tokens within the LLM's embedding space.

**Determining the Alignment Target.** First, we compute the target $L_2$ norm, $T$, by averaging the norms of all non-zero vectors from the language model's text embedding matrix, $\mathbf{W}_e$:

$$T = \frac{1}{|\mathcal{W}^*|} \sum_{\mathbf{w} \in \mathcal{W}^*} \|\mathbf{w}\|_2, \quad \text{where } \mathcal{W}^* = \{\mathbf{w} \in \mathbf{W}_e \mid \|\mathbf{w}\|_2 > \epsilon\}. \tag{5}$$

Based on this target, the gain parameter $\mathbf{g}$ is initialized to a scalar $g_{\text{init}} = T/\sqrt{D}$.

**The Optimization Dilemma.** A critical challenge arises from the intrinsic properties of the LLM's embedding space. Specifically, pre-trained text embeddings exhibit an **extremely small magnitude** (e.g., $\|\mathbf{w}\|_2 \approx 1$ even for $D = 4096$). Consequently, aligning to this target requires initializing $\mathbf{g}$ to a minute value (e.g., $g_{\text{init}} \approx 0.01$). In standard backpropagation, the gradient flowing back to the vision encoder is scaled by this weight: $\nabla_{\hat{\mathbf{x}}}\mathcal{L} = \nabla_{\mathbf{y}}\mathcal{L} \odot \mathbf{g}$. Thus, a minute initialization triggers a *vanishing gradient problem*, detaching the vision encoder from supervision.

**Resolution via Global Weight Compensation.** To resolve this, we propose a **Global Weight Compensation** mechanism implemented via a backward hook. Instead of redefining the gradient simply as identity, we actively rescale the gradients to counteract the dampening effect of the small initialization. Formally, let $\bar{g} = \frac{1}{D}\sum_i |g_i|$ be the mean magnitude of the gain vector. We define the backward pass dynamics as:

$$\text{Backward}(\nabla_{\hat{\mathbf{x}}}\mathcal{L}) = \underbrace{(\nabla_{\mathbf{y}}\mathcal{L} \odot \mathbf{g})}_{\text{Standard Gradient}} \times \underbrace{\frac{1}{\bar{g}}}_{\text{Compensation Factor}} . \tag{6}$$

By multiplying the gradient by the inverse of the global weight magnitude, we effectively cancel out the scaling term ($\mathbf{g} \cdot \bar{g}^{-1} \approx 1$), restoring the gradient flow to a unit scale while maintaining the precise architectural norm alignment in the forward pass.

## 5.2 EXPERIMENTAL SETUP

Our experiments are conducted within the LLaVA-1.5 architectural framework. Specifically, we employ Llama-3.2-3B-Instruct (Grattafiori et al., 2024) and Qwen2.5-7B-Instruct (Bai et al., 2025) as the base language model and SigLIP-SO400M-Patch14-384 as the vision encoder. Further details are provided in Appendix E.

A detailed list of the evaluation benchmarks is provided in the Appendix E; for all tasks, we employed a greedy decoding strategy.

## 5.3 RESULTS AND ANALYSIS

### 5.3.1 MAIN PERFORMANCE GAINS

The results in Table 4 reveal a backbone-dependent response. For Llama-3.2, simple norm alignment suffices to yield robust improvements. Conversely, Qwen2.5 exhibits the predicted *vanishing gradient* pathology: the minute initialization stifles gradient flow to the adapter, causing naive alignment to improve only text metrics while stagnating on multimodal tasks. Introducing our Global Weight Compensation (GWC) resolves this optimization bottleneck, unlocking gains across both domains. While this validates that norm alignment is fundamental for holistic model capability, the potential for gradient oscillation with GWC suggests that exploring more stable compression strategies remains a vital direction for future research.

We visualized the attention matrices in Appendix G. The analysis reveals that in the baseline model, text-to-image attention is inappropriately and broadly concentrated on the bottom regions of the image. This suggests a failure in semantic fusion, caused by the positional proximity bias introduced by RoPE's distance-decay property. In stark contrast, our norm-aligned model's text-to-image attention correctly converges on the specific image regions that are semantically relevant to the text query. This visual evidence provides direct confirmation that our method successfully restores meaningful cross-modal attention by correcting the underlying dynamic imbalance, thus enabling true feature fusion.

We also analyze the temporal evolution of layer-wise similarity and its convergence behavior during pre-training in Appendix F.

### 5.3.2 ABLATION STUDY: THE CRITICAL ROLE OF INITIALIZATION

To isolate the effect of our proposed initialization strategy, we conducted a crucial ablation study. We compared our method against a baseline where the added LayerNorm layer was initialized with default parameters (gain=1, bias=0). We analyzed the learned parameters immediately after the

Table 4: Performance comparison on various benchmarks across different backbones. **w/ Norm (w/o GWC)** denotes norm alignment with naive initialization (leading to gradient vanishing). **w/ Norm (w/ GWC)** denotes our proposed **Global Weight Compensation** mechanism.

| Model | Method | MMBench$_{dev}$ | MM-Star | POPE | SEED-Bench-2 | OCRBench |
|-------|--------|-----------------|---------|------|--------------|----------|
| **Llama** | w/o Norm | 71.39 | 37.72 | 88.14 | 42.86 | 40.70 |
| | w/ Norm (w/o GWC) | **72.16** (+0.77) | 41.19 (+3.47) | **88.88** (+0.74) | **47.26** (+4.40) | **45.60** (+4.90) |
| | **w/ Norm (w/ GWC)** | 71.82 (+0.43) | **41.24** (+3.52) | 88.26 (+0.12) | 45.56 (+2.70) | 44.10 (+3.40) |
| **Qwen** | w/o Norm | 76.80 | 50.34 | 87.51 | 56.65 | 47.00 |
| | w/ Norm (w/o GWC) | 75.60 (-1.20) | 48.08 (-2.26) | 87.83 (+0.32) | **59.51** (+2.86) | 47.60 (+0.60) |
| | **w/ Norm (w/ GWC)** | **77.66** (+0.86) | **50.58** (+0.24) | **87.87** (+0.36) | 58.27 (+1.62) | **49.40** (+2.40) |

| Model | Method | ScienceQA | AI2D | HellaSwag | MMLU | Avg |
|-------|--------|-----------|------|-----------|------|-----|
| **Llama** | w/o Norm | 78.99 | 60.17 | 65.96 | 45.19 | 59.01 |
| | w/ Norm (w/o GWC) | 80.83 (+1.84) | **63.24** (+3.07) | **66.01** (+0.05) | **53.21** (+8.02) | **62.04** (+3.03) |
| | **w/ Norm (w/ GWC)** | **81.00** (+2.01) | 61.85 (+1.68) | 65.99 (+0.03) | 51.60 (+6.41) | 61.27 (+2.26) |
| **Qwen** | w/o Norm | 82.81 | 73.74 | 70.56 | 71.02 | 68.49 |
| | w/ Norm (w/o GWC) | 82.20 (-0.61) | 72.70 (-1.04) | **73.73** (+3.17) | 71.14 (+0.12) | 68.71 (+0.22) |
| | **w/ Norm (w/ GWC)** | **82.93** (+0.12) | **74.61** (+0.87) | 71.64 (+1.09) | **71.74** (+0.72) | **69.41** (+0.92) |

**LLaVA Stage 1 pre-training phase.** As shown in Table 5, the parameters of the default-initialized layer remained largely unchanged from their initial state, indicating that the optimization process failed to begin effectively without a reasonable starting point. In contrast, our method shows meaningful parameter updates even after this initial stage. This demonstrates that simply adding a norm layer is insufficient; our targeted initialization is essential to place the parameters in a gradient-rich region of the loss landscape, enabling effective learning.

Table 5: Learned parameters of the added LayerNorm layer after Stage 1 pre-training, comparing default initialization with our proposed strategy.

| Parameter | Metric | Default Init (After Stage 1) | Our Init (After Stage 1) |
|-----------|--------|------------------------------|--------------------------|
| **Gain (g)** | $L_2$ Norm | 53.2500 | 2.2812 |
| | Mean of Abs. | 0.9609 ($\pm$ 0.0005) | 0.0400 ($\pm$ 0.0001) |
| **Bias ($\beta$)** | Mean of Abs. | 0.0175 ($\pm$ 0.0002) | 0.0152 ($\pm$ 0.0001) |

### 5.3.3 DIAGNOSTIC ANALYSIS: VERIFYING THE MECHANISM OF IMPROVEMENT

Finally, we performed a diagnostic analysis to investigate whether the performance gains correlate with the mitigation of the dynamic imbalance we identified. We analyzed the internal states of the fully trained model (after Stage 2) comparing the baseline against our norm-aligned method. Figure 4 visualizes two key metrics:

- **Layer-wise L2 Norms (Fig. 4a):** The left panel indicates that our method effectively harmonizes the visual token norms with the text token norms starting from the initial layers and maintains this alignment throughout the model's depth. In contrast, the baseline model exhibits a marked and persistent norm divergence.

- **Inter-layer Cosine Similarity (Fig. 4b):** The right panel illustrates the corresponding shift in update dynamics. With our intervention, the update rates (proxied by cosine similarity) of visual and text tokens exhibit significantly improved synchronization. This observation suggests that our method successfully mitigates the extreme asymmetry present in the baseline, where the persistently high similarity of visual tokens pointed to substantial "representational inertia."

Collectively, these findings corroborate our theoretical framework: correcting the static norm disparity appears to alleviate the asymmetric update rates, thereby facilitating the observed improvements in downstream performance.

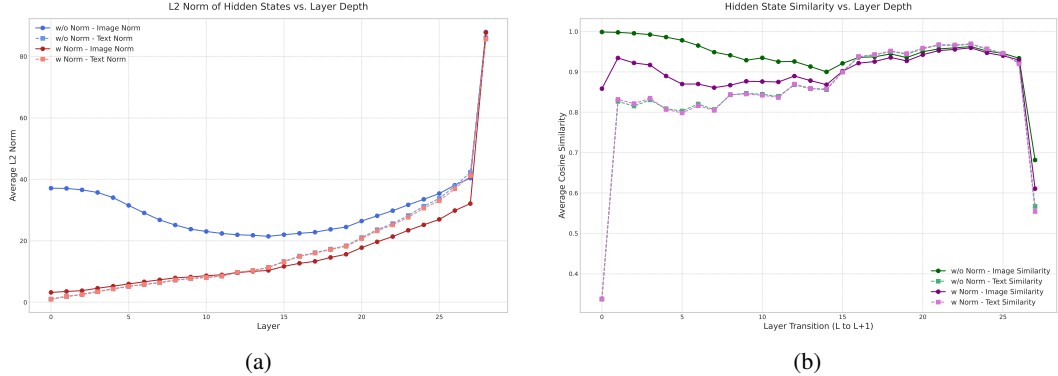

(a)                (b)

Figure 4: A comparison of token dynamics with and without our norm alignment method. (a) shows the layer-wise L2 norm evolution, while (b) shows the inter-layer cosine similarity, which acts as a proxy for update rate.

## 6 CONCLUSION

Our analysis reveals a critical, previously undiscovered dynamic within Pre-Norm MLLMs: an "asymmetric update." We have formalized this dynamic theoretically and validated it empirically, showing it to be a direct consequence of the severe norm disparity between visual and text tokens. This analysis demonstrates that the dynamic manifests as "representational inertia" in high-norm visual tokens, fundamentally impairing cross-modal fusion at an architectural level. It was this deep analysis of the mechanism that motivated our targeted solution of enforcing norm alignment via a single LayerNorm with global Weight Compensation. The resulting significant performance gains on both multimodal and, critically, text-only tasks, serve as compelling validation for our core analysis, confirming that resolving this dynamic imbalance unlocks the model's full potential.

## 7 ACKNOWLEDGEMENT

This work is supported by National Natural Science Foundation of China (92470121, 62402016), National Key R&D Program of China (2024YFA1014003), Zhongguancun Academy (Grant No.s C20250204, C20250602), and High-performance Computing Platform of Peking University.

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

## A BACKGROUND & RELATED WORK

### A.1 MULTIMODAL LARGE LANGUAGE MODELS

The remarkable success and emergent capabilities of Large Language Models (LLMs) in natural language processing have catalyzed efforts to generalize their powerful abilities to other modalities (Achiam et al., 2023; Hurst et al., 2024; Comanici et al., 2025; Fu et al., 2025; Bai et al., 2025; Yang et al., 2025; Wu et al., 2024; Chen et al., 2025). In the multimodal domain, this trend has spurred the rapid development of Multimodal Large Language Models (MLLMs).

Early explorations in MLLMs, such as Flamingo (Alayrac et al., 2022) and BLIP-2 (Li et al., 2023a), primarily relied on the cross-attention mechanism for modality fusion. A subsequent evolution witnessed a paradigm shift toward a simpler and more efficient approach, an approach popularized by LLaVA (Liu et al., 2023) that has now become the undisputed mainstream. The LLaVA-style architecture eschews the complexity of cross-attention in favor of a more direct solution: it employs a simple projection module, typically a Multi-Layer Perceptron (MLP), to map visual token features directly into the LLM's word embedding space. Conceptually, this treats the image as a sequence of special "visual words" prepended to the text input, which are then processed uniformly by the LLM in an auto-regressive manner. The simplicity, scalability, and powerful performance of this paradigm—particularly when combined with visual instruction tuning—have firmly established it as the foundational blueprint for the vast majority of today's advanced MLLMs

Despite the dominance of the LLaVA paradigm, the pursuit of optimal cross-modal fusion remains an active area of research. Investigators continue to experiment with more sophisticated projector designs (Team et al., 2025; Hong et al., 2025; Cha et al., 2024), alternative representation schemes like visual vocabularies (Lu et al., 2024), or deeper fusion strategies (Meng et al., 2024), novel methods for adapting the core architectures of large models for multimodal scenarios (Deng et al., 2025; Wei et al., 2025; Li & Zhang, 2025). Beyond these methods, other researchers have approached the challenge from a data-centric perspective (Bai et al., 2024; Liu et al., 2024c).

### A.2 NORMALIZATION

Normalization layers are a cornerstone of modern deep learning, designed to stabilize the training process and accelerate model convergence. By re-scaling the distribution of activations between layers, normalization effectively mitigates the internal covariate shift problem and ensures smooth gradient propagation in deep networks. While Batch Normalization (BN) (Ioffe & Szegedy, 2015) was a seminal work in this area, its dependency on batch size makes it less suitable for natural language processing tasks with variable sequence lengths. Layer Normalization (LayerNorm) was therefore introduced, performing normalization along the feature dimension independently of the batch, and it quickly became the standard for Transformer architectures (Vaswani et al., 2017). This paradigm was further refined by RMSNorm Zhang & Sennrich (2019), which improves computational efficiency by removing the mean re-centering step while maintaining performance, leading to its widespread adoption in many modern LLMs such as Llama.

A critical design axis in Transformer architectures is the placement of the normalization layer relative to the residual connection, giving rise to the Pre-Norm and Post-Norm paradigms Xiong et al. (2020). The original Post-Norm design applies normalization after the residual addition, which can help preserve strong representational fidelity but is often prone to training instability in deep models. In contrast, the Pre-Norm approach places normalization within the residual branch, greatly improving gradient flow and training stability by maintaining a "clean" skip-connection path. This has made it the de facto standard for large-scale language models. However, the Pre-Norm architecture has a well-documented side effect: because the hidden states on the main path are never re-normalized, their L2 norm tends to accumulate and grow with network depth (Zhuo et al., 2025).

Qi et al. (2025) have also observed the norm discrepancy between visual and textual tokens in MLLMs, they predominantly attribute the resulting performance degradation to the failure of Rotary Positional Embeddings (RoPE (Su et al., 2024) ) in handling high-magnitude features. However, we argue that this attribution overlooks the fundamental mechanics of the Pre-Norm architecture. Specifically, in this paradigm, normalization is applied before the query and key projections; consequently, the vectors processed by RoPE are already normalized.

Recently, the community has begun to re-evaluate this classic dichotomy, spurring research into alternative placement strategies. Recent works, like Peri-Norm (Kim et al., 2025) and Hybrid-Norm (Zhuo et al., 2025), have begun to explore combining normalization at different points of the residual connection to merge the benefits of both paradigms. These efforts, however, aim to find a universally optimal static design for unimodal models. In contrast, our work takes a diagnostic perspective: rather than proposing a new general architecture, we are the first to deeply analyze and reveal how the de facto standard Pre-Norm design itself directly induces a destructive dynamic imbalance within the multimodal context.

# B  APPENDIX: DETAILED DERIVATION AND PROOFS

This appendix provides the full mathematical derivation for the claims made in Section 3.3. We connect the architectural cause (norm imbalance) to the functional failure (SNR collapse) through a continuous four-step theoretical framework.

## B.1  STEP 1: KINEMATICS — NORM IMBALANCE INDUCES VELOCITY ASYMMETRY

We begin by establishing the kinematic relationship between token norm and update speed. Consider a hidden state $\mathbf{h}$ updated by a residual vector $\Delta\mathbf{h}$. The update can be decomposed into a parallel component (magnitude scaling) and an orthogonal component (direction rotation). The effective angular velocity, $\theta_{\text{eff}}$, is determined by the ratio of the orthogonal update to the current state magnitude.

Under the assumptions of **Uniform Update Magnitude** ($\|\Delta\mathbf{h}\| \approx C$) and **Consistent Update Geometry**, the tangent of the effective rotation angle is derived as:

$$\tan(\theta_{\text{eff}}) = \frac{C\sin(\phi)}{\|\mathbf{h}\|_2 + C\cos(\phi)} \tag{7}$$

This formula establishes an inverse relationship: **higher norm implies lower angular velocity**. Applying this to the visual ($\mathbf{h}_{\text{vis}}$) and text ($\mathbf{h}_{\text{txt}}$) tokens, where $\|\mathbf{h}_{\text{vis}}\| \gg \|\mathbf{h}_{\text{txt}}\|$, we arrive at a fundamental velocity asymmetry:

$$\theta_{\text{vis}} \ll \theta_{\text{txt}} \tag{8}$$

**Transition:** Having established that the modalities rotate at significantly different speeds, we next model how this velocity mismatch affects the evolution of their alignment over time.

## B.2  STEP 2: DYNAMICS — RECURSIVE DECAY OF SEMANTIC SIMILARITY

We now derive the law governing the evolution of the cosine similarity between the two modalities across layers. Let $\Theta^{(l)}$ be the angle between the visual and text hidden states at layer $l$.

**Theorem 1: Recursive Decay.**  Assuming the rotational updates are drawn from a symmetric distribution in the orthogonal subspace, the expected cosine similarity evolves according to:

$$\mathbb{E}[\cos(\Theta^{(l+1)}) \mid \Theta^{(l)}] = \gamma_{\text{eff}}^{(l)} \cdot \cos(\Theta^{(l)}) \tag{9}$$

where the retention factor $\gamma_{\text{eff}}^{(l)}$ is the product of the cosines of the individual angular velocities:

$$\gamma_{\text{eff}}^{(l)} = \cos(\theta_{\text{vis}}^{(l)})\cos(\theta_{\text{txt}}^{(l)}) \tag{10}$$

**Transition:** This theorem tells us that alignment retention depends on the product $\cos(\theta_{\text{vis}})\cos(\theta_{\text{txt}})$. The critical question is: Does the extreme asymmetry derived in Step 1 make this retention factor better or worse compared to a balanced scenario?

## B.3  STEP 3: OPTIMIZATION — ASYMMETRY ACCELERATES GEOMETRIC DIVERGENCE

We compare the decay factor $\gamma$ in the Imbalanced case ($\theta_{\text{vis}} \ll \theta_{\text{txt}}$) against the Balanced case ($\theta_{\text{vis}} \approx \theta_{\text{txt}}$).

**Lemma 1** (Asymmetry Maximizes Decay Rate). *For a fixed total update potential (represented by the geometric mean of angular velocities), the similarity retention factor $\gamma = \cos(\theta_1)\cos(\theta_2)$ is maximized when velocities are symmetric ($\theta_1 = \theta_2$). Conversely, asymmetry strictly decreases $\gamma$.*

*Proof.* Maximizing $\gamma$ is equivalent to minimizing $1/\gamma^2 = (1 + \tan^2\theta_1)(1 + \tan^2\theta_2)$. By the AM-GM inequality, for a fixed product of tangents, the sum $\tan^2\theta_1 + \tan^2\theta_2$ is minimized when $\tan\theta_1 = \tan\theta_2$. Thus, the asymmetric condition $\|\mathbf{h}_{\text{vis}}\| \gg \|\mathbf{h}_{\text{txt}}\|$ forces the system into a suboptimal regime where similarity decays strictly faster than in the balanced case. $\square$

**Definition of the Lag Angle $\Delta\phi$.** Since the similarity decays faster in the imbalanced case, the final angle between the vectors will be strictly larger. We define this accumulated geometric deficit as the **Lag Angle**:

$$\Delta\phi = \Theta_{\text{imb}} - \Theta_{\text{bal}} > 0 \tag{11}$$

**Transition:** We have proven that visual tokens lag behind the optimal alignment direction by $\Delta\phi$. Finally, we must determine if the attention mechanism can tolerate this lag or if it leads to functional failure.

### B.4 STEP 4: FROM GEOMETRIC MISALIGNMENT TO SIGNAL-TO-NOISE RATIO COLLAPSE

Finally, we connect the geometric lag $\Delta\phi$ derived in Step 3 to the failure of the attention mechanism. We analyze the interaction between the learnable projection matrices $\mathbf{W} = \mathbf{W}_Q^T\mathbf{W}_K$ and the degraded input statistics.

**1. Spectral Degradation of the Semantic Signal.** The expected attention score is given by the bilinear form $\mathbb{E}[S] = \text{Tr}(\mathbf{W}\mathbf{C}_{vu})$, where $\mathbf{C}_{vu} = \mathbb{E}[\mathbf{v}\mathbf{u}^T]$ is the cross-covariance matrix of the inputs. In the Imbalanced case, the geometric lag $\Delta\phi$ causes the visual vector $\mathbf{v}_{\text{imb}}$ to deviate from the optimal alignment direction. By performing an orthogonal decomposition along the signal direction (aligned with the text vector $\mathbf{u}$), we have:

$$\mathbf{v}_{\text{imb}} = \cos(\Delta\phi)\mathbf{v}_{\text{signal}} + \sin(\Delta\phi)\mathbf{v}_{\perp} \tag{12}$$

Since the orthogonal component $\mathbf{v}_{\perp}$ contains no correlation with the query, the singular values (spectral energy) of the covariance matrix are dampened:

$$\mathbf{C}_{\text{imb}} \approx \cos(\Delta\phi)\mathbf{C}_{\text{bal}} \tag{13}$$

**2. The Fundamental Spectral Upper Bound.** We consider the maximization of the attention score under the constraint that the model parameters must remain finite. Let the magnitude of the projection matrix be bounded by some fixed radius $R$ (i.e., $\|\mathbf{W}\|_F \leq R$). We invoke **Von Neumann's Trace Inequality**, which states that the trace of a matrix product is bounded by the dot product of their singular values: $\text{Tr}(\mathbf{W}\mathbf{C}) \leq \sum \sigma_i(\mathbf{W})\sigma_i(\mathbf{C})$. Applying this, the maximum achievable expected score is strictly limited by the spectrum of the input covariance:

$$\max_{\|\mathbf{W}\|_F \leq R} \mathbb{E}[S_{\text{rel}}^{\text{imb}}] \leq R \sum \sigma_i(\mathbf{C}_{\text{imb}}) \approx \cos(\Delta\phi)\left(R \sum \sigma_i(\mathbf{C}_{\text{bal}})\right) \tag{14}$$

Since $\Delta\phi > 0$, we have $\cos(\Delta\phi) < 1$. This proves a structural theorem: **For any fixed capacity $R$ of the projection layers, the norm-induced geometric lag strictly reduces the maximum extractable attention score.** The model cannot recover the lost signal energy without unboundedly increasing its weights.

**3. Signal-to-Noise Ratio (SNR) Collapse.** The attention mechanism's efficacy relies on the gap between signal and noise:

$$\text{Gap}_{\text{imb}} = \mathbb{E}[S_{\text{rel}}^{\text{imb}}] - \mathbb{E}[S_{\text{irrel}}^{\text{imb}}] \tag{15}$$

- **Signal:** As proven above, the maximum signal is capped by the factor $\cos(\Delta\phi)$.
- **Noise:** For irrelevant tokens, the vectors are initialized to be approximately orthogonal ($\Theta_{\text{irrel}} \approx 90°$). The geometric divergence acts as an isotropic random walk, introducing no systematic bias. Thus, the noise floor remains constant: $\mathbb{E}[S_{\text{irrel}}^{\text{imb}}] \approx 0$.

Consequently, the Signal-to-Noise Ratio collapses ($\text{Gap}_{\text{imb}} < \text{Gap}_{\text{bal}}$). This reduced gap forces the Softmax function to produce a higher-entropy, more uniform distribution, mathematically explaining the diffuse attention maps observed in experiments.

## C  EMPIRICAL ANALYSIS OF EMBEDDING NORM SHIFTS

To investigate the impact of multimodal training on the embedding space, we analyzed the statistical properties of token embeddings across the Qwen2 and Qwen2.5 model families. We systematically compared the pre-trained pure language models (Base) against their visual-language counterparts (VL).

### C.1  GLOBAL STATISTICS

Table 6 presents the global statistics (Mean and Standard Deviation) of the embedding norms. We observe that while Qwen2-VL maintains a mean norm similar to its base model, Qwen2.5-VL exhibits a noticeable systemic inflation (Mean: $0.80 \rightarrow 0.90$), indicating a broader shift in the embedding distribution during multimodal alignment.

Table 6: Comparison of global embedding statistics ($L_2$ Norm Mean and Standard Deviation) between Base (Pure Text) and VL (Multimodal) models.

| Metric | Qwen2 Family (7B) | | Qwen2.5 Family (7B) | |
|---|---|---|---|---|
| | Base | VL | Base | VL |
| Mean Norm ($\mu$) | 0.6908 | 0.6820 | 0.8031 | 0.8965 |
| Std. Deviation ($\sigma$) | 0.1610 | 0.1588 | 0.1801 | 0.1821 |

### C.2  DETAILED ANALYSIS OF TOP-NORM TOKENS

Tables 7 and 8 detail the Top-10 tokens with the highest $L_2$ norms. The comparison reveals a fundamental structural change in the embedding space:

1. **Base Models (Pure Text):** In pure language models, the tokens with the highest norms are typically rare subwords or specific syntactic markers (e.g., `'áveis'`, `'"=>'`). The maximum norm is relatively contained ($\approx 1.0 - 1.1$).

2. **VL Models (Multimodal):** Upon multimodal training, the visual boundary tokens (`<|vision_start|>`, `<|vision_end|>`) emerge as extreme outliers. In Qwen2-VL, they reach norms of $\approx 2.4$, far exceeding the previous maximums. This explicitly confirms that the model allocates significantly larger magnitudes to visual anchors to accommodate the high-norm visual features.

Table 7: **Qwen2 Family Comparison:** Top-10 tokens by $L_2$ norm. Note how the visual tokens (in VL) far exceed the magnitude of the outliers in the Base model.

| Rank | Qwen2-7B-Instruct (Base) | | Qwen2-VL-7B-Instruct (VL) | |
|---|---|---|---|---|
| | Token | $L_2$ Norm | Token | $L_2$ Norm |
| 1 | `'áveis'` | 1.0273 | `<|vision_start|>` | **2.4590** |
| 2 | `'"=>'` | 0.9741 | `<|vision_end|>` | **2.3320** |
| 3 | `>();\\n\\n` | 0.9258 | `'áveis'` | 1.0049 |
| 4 | `'s'` | 0.9072 | `'"=>'` | 0.9458 |
| 5 | `'is'` | 0.9004 | `'s'` | 0.9067 |
| 6 | `'an'` | 0.8999 | `>();\\n\\n` | 0.9009 |
| 7 | `'le'` | 0.8994 | `' .'` | 0.8999 |
| 8 | `'ed'` | 0.8979 | `'is'` | 0.8989 |
| 9 | `' .'` | 0.8965 | `'an'` | 0.8975 |
| 10 | `'ar'` | 0.8965 | `'le'` | 0.8965 |

## D  DETAILED IMPLEMENTATION OF NORM ALIGNMENT

Our norm alignment layer, denoted as `GlobalWeightCompensatedLayerNorm`, is designed to enforce norm compression in the forward pass while preserving gradient magnitude in the back-

Table 8: **Qwen2.5 Family Comparison:** Top-10 tokens by $L_2$ norm. In the Base model, visual tokens are initialized to zero or unused. In the VL model, they become the largest vectors.

| | Qwen2.5-7B-Instruct (Base) | | Qwen2.5-VL-7B-Instruct (VL) | |
|---|---|---|---|---|
| **Rank** | **Token** | **$L_2$ Norm** | **Token** | **$L_2$ Norm** |
| 1 | `'áveis'` | 1.1201 | `<|vision_end|>` | **2.1523** |
| 2 | `','` | 1.1025 | `<|vision_start|>` | **1.4004** |
| 3 | `'is'` | 1.0322 | `','` | 1.2764 |
| 4 | `'s'` | 1.0303 | `' unb'` | 1.2158 |
| 5 | `'an'` | 1.0293 | `'áveis'` | 1.1748 |
| 6 | `'en'` | 1.0283 | `':\\n\\n'` | 1.1670 |
| 7 | `'(Chinese Char)'` | 1.0264 | `'en'` | 1.1670 |
| 8 | `' out'` | 1.0264 | `'s'` | 1.1650 |
| 9 | `'on'` | 1.0254 | `'(Chinese Char)'` | 1.1650 |
| 10 | `'le'` | 1.0254 | `'(Chinese Char)'` | 1.1650 |

ward pass. Unlike standard LayerNorm, we introduce a gradient compensation mechanism to handle the extremely small initialization of the gain parameter.

**Forward Pass (Standard LayerNorm).** Given the input vector $\mathbf{x} \in \mathbb{R}^D$ (representing the projected visual tokens), we first compute the mean $\mu$ and variance $\sigma^2$ across the feature dimension:

$$\mu = \frac{1}{D} \sum_{i=1}^{D} x_i, \quad \sigma^2 = \frac{1}{D} \sum_{i=1}^{D} (x_i - \mu)^2. \tag{16}$$

The input is then normalized and transformed by the affine parameters, the learnable gain $\mathbf{g}$ and bias $\boldsymbol{\beta}$:

$$\hat{\mathbf{x}} = \frac{\mathbf{x} - \mu}{\sqrt{\sigma^2 + \epsilon}}, \quad \mathbf{y} = \hat{\mathbf{x}} \odot \mathbf{g} + \boldsymbol{\beta}. \tag{17}$$

Crucially, consistent with our method description in Section 5.1, $\mathbf{g}$ is initialized to a small scalar value (derived from the target text norm $T$) to ensure immediate alignment, while $\boldsymbol{\beta}$ is initialized to zero.

**Backward Gradient Compensation.** Since $\mathbf{g}$ is initialized to a minute value, standard backpropagation would attenuate the gradients flowing back to the input $\hat{\mathbf{x}}$ (and consequently to the vision encoder) by a factor proportional to $\|\mathbf{g}\|$. To prevent this, we register a backward hook on the normalized tensor $\hat{\mathbf{x}}$ to dynamically rescale the gradients.

The compensation process proceeds as follows during training:

1. **Global Gain Aggregation:** We compute the mean absolute value of the gain parameter vector $\mathbf{g}$ to obtain a global scaling scalar:

$$\mu_g = \frac{1}{D} \sum_{i=1}^{D} |g_i|. \tag{18}$$

2. **Safety Clamping:** To ensure numerical stability and prevent division by zero (in the rare event of parameter collapse), we clamp the scalar with a minimal threshold $\delta = 10^{-3}$:

$$\mu_{\text{safe}} = \max(\mu_g, \delta). \tag{19}$$

3. **Compensation Factor Application:** The gradient $\nabla_{\hat{\mathbf{x}}} \mathcal{L}$ flowing backwards through the normalization operation is scaled by the inverse of this scalar:

$$\nabla_{\hat{\mathbf{x}}} \mathcal{L}_{\text{scaled}} = \nabla_{\hat{\mathbf{x}}} \mathcal{L} \times \frac{1}{\mu_{\text{safe}}}. \tag{20}$$

This mechanism effectively decouples the forward scale (controlled by $\mathbf{g}$) from the backward gradient scale (restored to $\approx 1.0$). By preserving the gradient magnitude, we ensure that the vision encoder receives effective supervision signals from the very first training iteration, despite the severe norm compression applied at the interface.

# E    TRAINING DETAILS

## E.1    EXPERIMENTAL SETUP

Our experiments are conducted within the LLaVA-1.5 architectural framework. To ensure a comprehensive evaluation, we employ two distinct base language models: Llama-3.2-3B-Instruct and Qwen2.5-7B-Instruct. Both models are coupled with SigLIP-SO400M-Patch14-384 as the vision encoder. We follow a unified two-stage training protocol for both backbones: the first stage consists of one epoch of feature alignment pre-training on the LLaVA-558K dataset, using a learning rate of 1e-3, a per-device batch size of 2, and 2 gradient accumulation steps, resulting in a global batch size of 256. This is followed by one epoch of full-model instruction tuning on the LLaVA-NeXT dataset, for which the learning rate is decreased to 1e-5 for the language model and 2e-6 for the vision encoder, with a per-device batch size of 1 and 4 gradient accumulation steps, corresponding to a global batch size of 128. Across both stages, we utilize a cosine learning rate scheduler with a warmup ratio of 0.03 and set weight decay to 0. Notably, we do not employ dynamic high-resolution strategies; all images are uniformly resized to $384 \times 384$. To ensure reproducibility, we set the random seed to 42 for all experiments.

To comprehensively evaluate the model's performance, we assessed its capabilities on both multimodal and text-only tasks. The model's multimodal abilities were benchmarked against a comprehensive suite of benchmarks, including MMBench-EN (Liu et al., 2024a), MM-Star (Chen et al., 2024b), OCRBench (Liu et al., 2024b), SEED-Bench-2-Plus (Li et al., 2024b), ScienceQA (Lu et al., 2022), AI2D (Kembhavi et al., 2016), and POPE (Li et al., 2023b). Furthermore, to gauge its core language understanding and commonsense reasoning skills, we evaluated its performance on the HellaSwag (Zellers et al., 2019) and MMLU (Hendrycks et al., 2020) benchmarks.

# F    TRAINING DYNAMICS

To further probe the temporal evolution of these dynamics, we tracked the layer-wise cosine similarity of image and text tokens across varying checkpoints during the multimodal pre-training phase (Figure 5a) and 5b). As shown in Figure 5a, without the additional normalization layer, visual tokens exhibit persistently high inter-layer similarity. This pattern remains virtually static throughout the training process, confirming that the "representational inertia" induced by norm discrepancy is a persistent barrier, effectively locking visual features against semantic transformation from the very beginning. In contrast, our method (Figure 5b) initiates with a significantly lower visual similarity, indicating active feature updates. Interestingly, as training progresses, we observe a gradual upward trend in similarity, which correlates with the optimization of the added LayerNorm's gain parameter (**g**). Crucially, as observed in the figure, this upward trend has converged. This phenomenon implies that perfectly consistent update rates between image and text tokens might not be optimal, suggesting that a certain degree of divergence between the two could be appropriate. However, this is fundamentally distinct from the significant update inconsistency driven by the massive initial norm discrepancy observed in the baseline.

# G    APPENDIX: ATTENTION VISUALIZATION

In each pair of heatmaps, the bottom image shows the model with norm applied, while the top image shows the baseline model. The caption for each pair corresponds to the text query used.

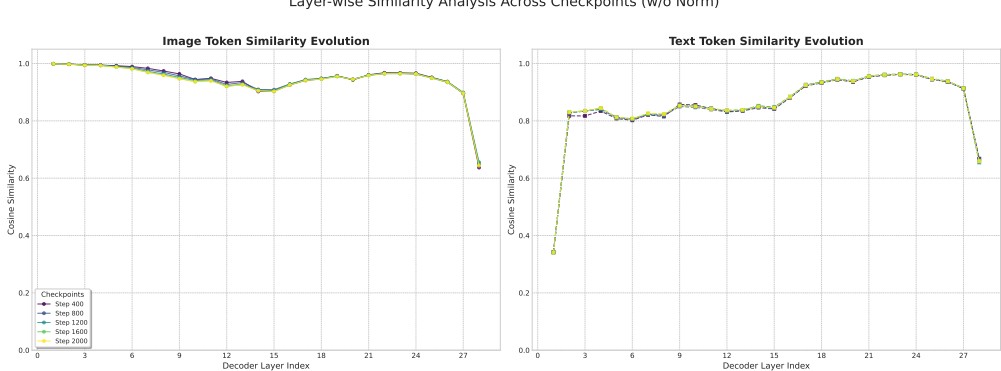

(a) Evolution of layer-wise cosine similarity across different checkpoints during the multimodal pre-training phase (w/o Norm).

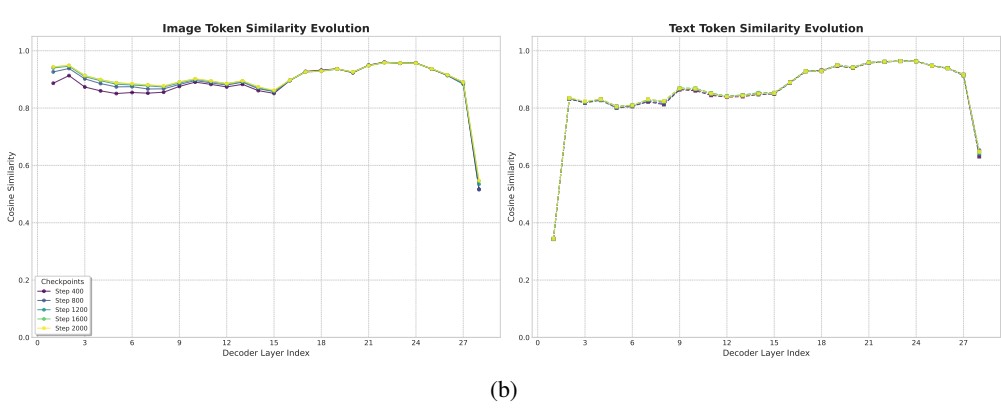

(b)

Figure 5: Evolution of layer-wise cosine similarity across different checkpoints during the multimodal pre-training phase (w Norm).

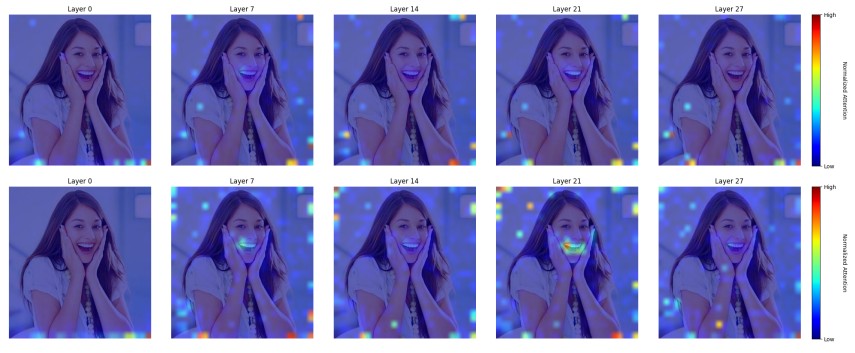

(a) Which mood does this image convey?

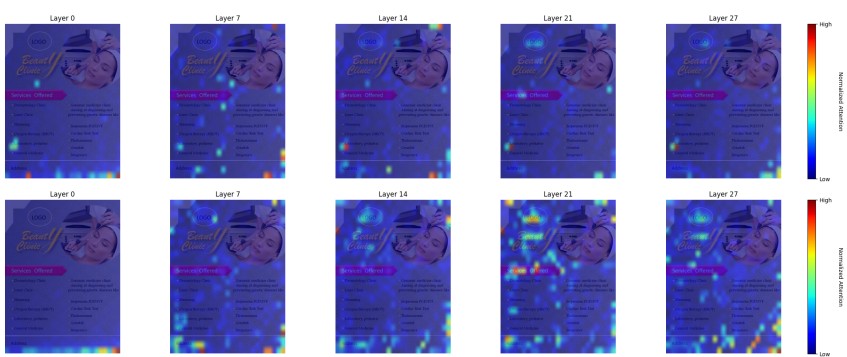

(b) What is the main subject of the flyer seen in the image?

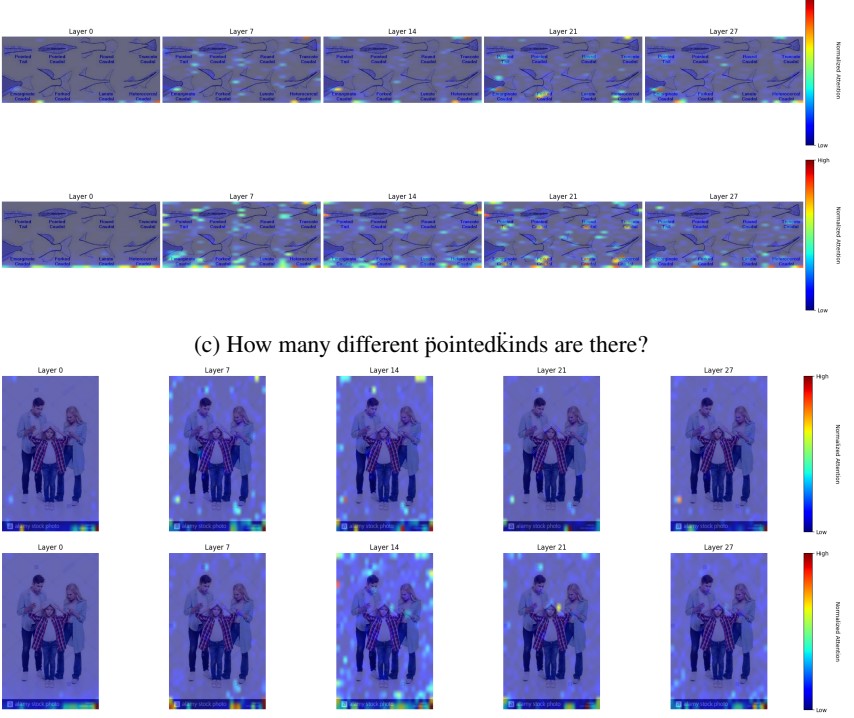

(c) How many different pointed kinds are there?

(d) What type of family is shown in the image

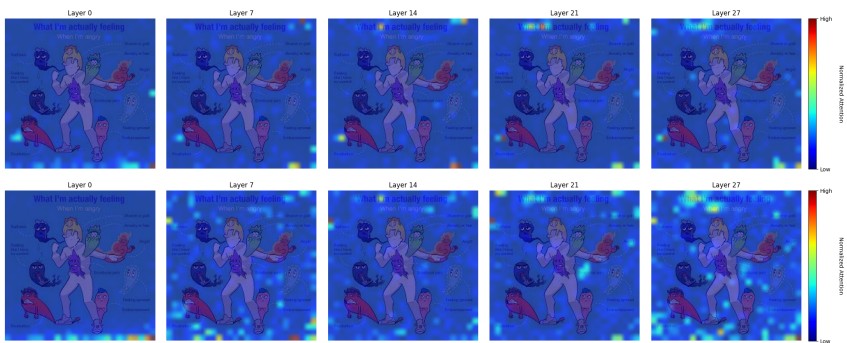

(a) What emotion is portrayed in this image?

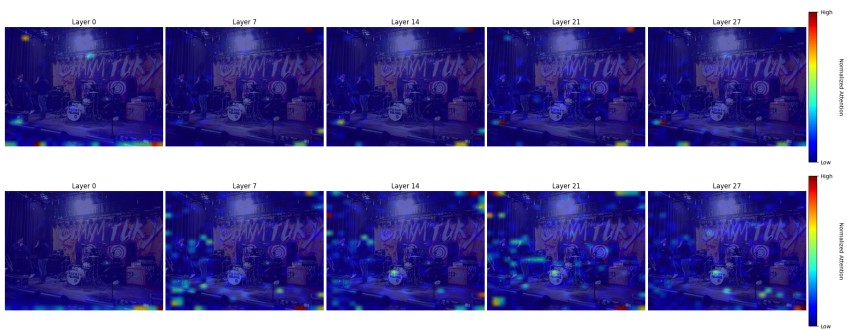

(b) How many people are performing on the stage?

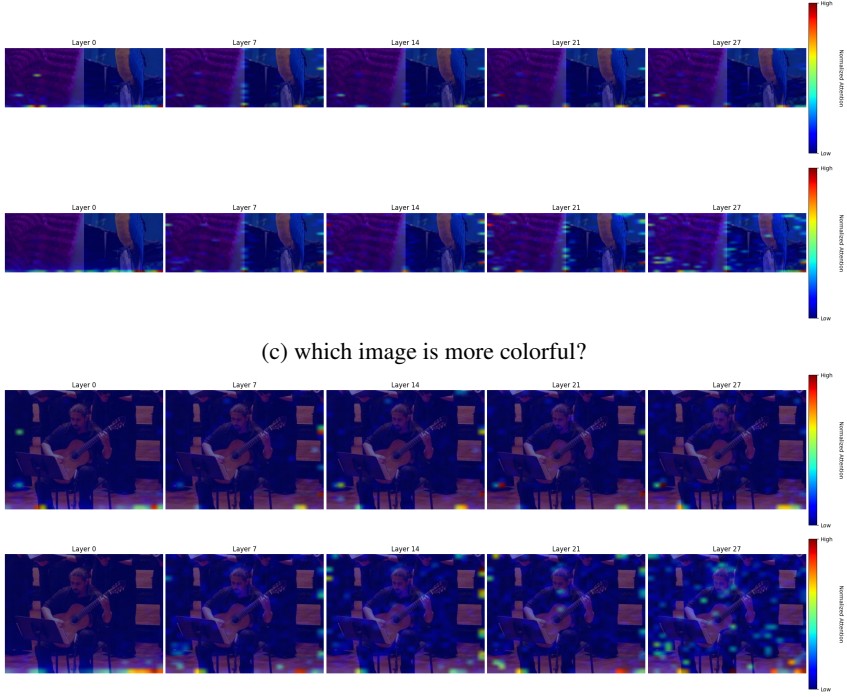

(c) which image is more colorful?

(d) What is the main focus of the image?

