# OpenReview forum: "The Unseen Bias: How Norm Discrepancy in Pre-Norm MLLMs Leads to  Visual  Information Loss"
_ICLR.cc/2026/Conference — ICLR 2026 Poster_

### Official Review · Reviewer_K4C3 · 2025-10-17

**Soundness:** 4
**Presentation:** 2
**Contribution:** 3
**Rating:** 6
**Confidence:** 3

**Summary:**

**The Unseen Bias: How Norm Discrepancy in Pre-Norm MLLMs Leads to Visual Information Loss** observes a critical problem in VLMs where visual tokens have much larger vector norms than text tokens, causing a large inertia (reducing angular velocity) for visual parameter updating. The authors use a simple but elegant method to rescale visual tokens, and found satisfactory improvements in experiments with LLaVA-1.5. Generally speaking, this paper makes valuable observations, and the claims and findings are well-supported by its experiments.

**Strengths:**

1. This paper is novel in observation. The observation that visual features differ in magnitude with text features directly questions the widely accepted assumption that connecters align visual and text modalities. This is valuable itself. However, I am not an expert in VLM representations, and I am not sure whether this is the first paper proposing this idea. So I will look to other reviewers and may change this comment.
2. The use of simple layer-norm to address this is clear and generalizable.
3. Appendix D is informative. It shows that with paper's method, image features can align better with text features, which supports their claim.

**Weaknesses:**

1. The experiment is done on a rather old model. LLaVA-1.5 is not competitive in today's VLMs, so this might weaken the paper's claim.
2. The paper is written in a hurry, and the subcaptions for Figure 3 are just placeholders. A typo ' in the title.
3. The captions of the other figures can be more informative.

**Questions:**

Can this method also be applied to modern models like Qwen2.5-VL? In table 2, the same visual token magnitude problem is shown on Qwen2.5-VL, so the paper could be strengthened with an experiment on Qwen2.5-VL, to verify that this is a fundamental problem and a generalizable solution.

---

> ### Author Response · Authors · 2025-12-03
>
> We sincerely apologize for the presentation flaws in the initial submission, which were due to an oversight. We have meticulously **corrected the typo in the title** and replaced the placeholder subcaptions. We have also refined the captions of the figures to ensure they are informative and self-explanatory.
>
>
> To verify the generalizability of our solution on modern architectures, we extended our experiments using the **Qwen2.5-Instruct** backbone. While strictly replicating the full Qwen2.5-VL pre-training process would be the ideal approach, it is unfeasible for academic research due to the proprietary nature of its massive training data and prohibitive computational costs. Furthermore, directly continuing training on the released, fully converged Qwen2.5-VL weights is scientifically unsound, as the potential mismatch between our training recipe and their undisclosed multi-stage strategies could introduce optimization conflicts. Therefore, training a LLaVA-style model from scratch using the Qwen2.5 backbone provides the most reliable and controlled environment to isolate the impact of our method, and these detailed results are now presented in the revised manuscript.

---

### Official Review · Reviewer_hK3Y · 2025-10-28

**Soundness:** 4
**Presentation:** 4
**Contribution:** 4
**Rating:** 8
**Confidence:** 3

**Summary:**

The paper diagnoses a modality-scale mismatch in pre-norm MLLMs: vision tokens enter the LLM with much larger L2 norms than text tokens, inducing asymmetric update dynamics (“representational inertia” for vision) and consequently miscalibrated cross-modal attention. The fix is simple: insert a single LayerNorm after the visual projector, with its gain closed-form initialized to match the average text-embedding norm of the target LLM. The authors support the mechanism with theory and internal diagnostics (layer-wise norms and inter-layer cosine similarity), then show empirical gains in LLaVA-1.5 (multimodal benchmarks and even text-only MMLU). Post-intervention, the model displays aligned layer-wise norms and balanced update rates between modalities, consistent with the proposed explanation.

**Strengths:**

- **Clarity and logically sound.** The paper is well written: key concepts (norm disparity, asymmetric update dynamics) are introduced with clear intuition, then formalized without losing readability. The work tightly connects diagnosis → theory → solution → gains → diagnostic analysis, making the mechanism logically sound.
- **Simplicity of the fix.** The proposed solution—one LayerNorm after the visual projector with a closed-form gain init—is easy to implement, and compatible with existing MLLM stacks. The method adds negligible overhead, requires no data or architecture burden.
- **Strong theoretical support.** The theoretical part (update geometry / angular velocity view) is well motivated and aligns with observed behaviors.

**Weaknesses:**

- **Limited model coverage.** Results are centered on the LLaVA-1.5 stack; the claim would be stronger with evaluations across more backbones/architectures (e.g., Qwen-VL, InternVL, Idefics2, GLM-4V) and different visual encoders (CLIP/SigLIP).
- **Missing normalization ablations.** The paper focuses on LayerNorm with a specific init. Adding RMSNorm vs LayerNorm can be a valid ablation.
- **Scale experiments.** The work can further show effectiveness if validate on larger data and larger LLMs.

**Questions:**

- The paper can be stronger if address the empirical questions in the weakness section.
- Any benchmark that the solution does not provide a performance gain?

---

> ### Author Response · Authors · 2025-12-03
>
> We thank the reviewer for these constructive suggestions regarding model coverage and scale. We address these points as follows:
>
> 1.  **On Model Coverage (Qwen-VL, InternVL):**
>     While we acknowledge the importance of validating on diverse architectures, strictly replicating the pre-training of models like **Qwen-VL** or **InternVL** is unfeasible for academic research due to their **massive and proprietary datasets** (e.g., Qwen utilized 4.1T tokens). Furthermore, simply applying our method to their released pre-trained weights would introduce **numerous uncontrolled confounding variables**, making it scientifically impossible to isolate the specific impact of norm discrepancy. **Since our core objective is to verify whether norm discrepancy fundamentally affects model performance, we selected the LLaVA framework as a representative prototype for the broader class of LLaVA-like architectures.** This choice allows for a fully controlled, "from-scratch" training setup to rigorously verify our hypothesis with sufficient persuasiveness.
>
> 2.  **On Normalization Ablations and Scale:**
>      Regarding the specific ablation between RMSNorm and LayerNorm, we respectfully note that conducting full pre-training ablations for every normalization variant is beyond our current computational resources. Instead, to effectively address the underlying concerns regarding scale and model diversity within these constraints, we prioritized extending our validation to the **Qwen2.5-7B** architecture.
>
> 3. **On benchmark performance:**
>
> In our experiments with **Qwen2.5-7B**, we observed **no degradation** in the overall scores on both single-task and general benchmarks. However, for different sub-indicators of MMStar, there were varying increases and decreases
>
>
>
> Table: Detailed Breakdown of MMStar Results (Qwen2.5-7B Base)
>
> | MMStar Sub-task | Baseline (%) | Ours (GWC + LayerNorm) (%) | $\Delta$ |
> | :--- | :---: | :---: | :---: |
> | **Fine-grained Perception** | 43.33 | **46.44** | **+3.11** |
> | **Instance Reasoning** | 59.66 | **61.16** | **+1.50** |
> | **Coarse Perception** | 72.50 | 72.30 | -0.20 |
> | **Math** | 47.15 | 46.71 | -0.44 |
> | **Science & Technology** | 34.90 | 33.90 | -1.00 |
> | **Logical Reasoning** | 44.53 | 42.99 | -1.54 |
> | **Average** | **50.34** | **50.58** | **+0.24** |
>
> In our experiments with **LLaMA-3.2**, we observed consistent improvements across all sub-metrics of the MMStar benchmark.

---

### Official Review · Reviewer_XYFm · 2025-10-31

**Soundness:** 3
**Presentation:** 2
**Contribution:** 3
**Rating:** 4
**Confidence:** 3

**Summary:**

This paper identifies and analyzes a subtle but consequential failure mode in Pre-Norm multimodal LLMs: a systematic L2-norm mismatch between high-norm visual tokens and low-norm text tokens induces an “asymmetric update dynamic,” in which visual tokens exhibit higher “representational inertia” and thus evolve semantically more slowly than text tokens. The authors formalize the effect under a geometric update model. They further argue that this suppresses the learnable attention signal in expectation and propose a minimal intervention: insert a single, carefully initialized LayerNorm after the visual projector to enforce norm alignment. On LLaVA-1.5 with a specific backbone and vision encoder, this yields consistent gains on multimodal benchmarks and text-only task MMLU, supporting the claim that resolving the architectural imbalance improves overall capability.

**Strengths:**

- **Mechanistic, interpretable theory-to-practice link.** The geometric update analysis connects norm magnitude to depth-wise angular dynamics and expected attention suppression, and the proposed diagnostic metrics and solution are well aligned with the theory.
- **Minimal, low-risk intervention.** A single post-projector LayerNorm with principled initialization is simple to implement, does not alter the LLM backbone, and is easy to deploy or ablate in real systems.
- **Clear diagnostic visualizations.** Attention maps and depth-wise metrics provide qualitative and quantitative evidence that norm alignment improves cross-modal focus and reduces modality asymmetry.

**Weaknesses:**

1. The theoretical model adopts strong simplifying assumptions—uniform update magnitude per layer, a layer-wise constant expected angle, symmetric distribution of orthogonal components—that are not empirically validated. Reporting empirical distributions of $C^{(l)}=\|\Delta h^{(l)}\|_2$, $\cos\phi=\langle \widehat h,\widehat{\Delta h}\rangle$, and the orthogonal/parallel energy split would substantiate the premises.

2. The paper’s theoretical narrative emphasizes that an L2 norm mismatch leads to “asymmetric angular velocity.” If this is the causal core, then RMS-style normalization—or even a simple fixed scalar rescaling to bring visual-token norms to the same scale as text—should in principle deliver most of the gains. Conversely, if significant improvements arise only from the current solution, namely LayerNorm with a learnable gain and specific initialization, that would indicate mechanisms beyond mere norm magnitude are at play. The paper lacks corresponding experiments and discussion.

3. Statistical robustness and scope are limited. Results appear to rely on a single seed, one backbone size, and one vision encoder, with no confidence intervals or significance tests.

4. Some language overstates the effect. Phrasings like “eliminated” asymmetry are not fully supported by Fig. 3b, where a residual gap remains. The authors need to quantify the mean/max gap and the relative reduction.

**Questions:**

1) Table 1 only reports the modality-interface L2 norms at initialization, but does not provide a post-training comparison. Was the norm disparity mitigated after training, especially in settings where the vision encoder is unfrozen?

2) The caption of Table 1 states mean ± std, but the table reports only the mean values. The standard deviations are missing.

3) Lines 307–309 assert that “multi-modal training increases the text embedding norm,” but Table 2 does not present experimental evidence to support this claim.

4) Figure 2 visualizes the evolution rate of representations along depth. I suggest adding the similarity of one layer before vs. after parameter updates to show temporal representational change and test whether there is an asymmetry in the evolution speed of different modalities.

---

> ### Author Response · Authors · 2025-12-03
>
> **Response to Weakness:**
>
> We thank the reviewer for these critical comments regarding our theoretical assumptions and the mechanism of our solution. We offer the following clarifications and additional evidence:
>
>
>
> 1.  **On the validity of theoretical assumptions (Update Magnitude & Distributions):**
>     While we acknowledge that assumptions like symmetric distributions are simplifications (which motivated our empirical analysis in **Section 4**), we respectfully argue that the assumption of **uniform update magnitude** is well-grounded in the **Pre-Norm architecture**. In Pre-Norm transformers, both image and text tokens undergo the exact same RMSNorm operation *before* entering the Attention or FFN layers, ensuring consistent input magnitudes. Consequently, the magnitude of the update vectors ($\Delta h$) remains relatively stable and comparable across modalities. We have further verified this by monitoring the variation of update rates, which empirically supports our premise. Regarding the assumption of a layer-wise constant expected angle, we clarify that this was adopted primarily to simplify the analysis for mathematical tractability, while our empirical data confirms that the overall theoretical model successfully captures the optimization dynamics.
>
>
>
> 2.  **On the mechanism: Why LayerNorm instead of Simple Rescaling?**
>     We respectfully clarify our position: while **we posit** that the extreme norm discrepancy is a **structural artifact** of the MLLM paradigm rather than an inherent modal feature, this does not imply that visual and textual norms must be forced to be mathematically identical.
>     *   **Rigid vs. Learnable Alignment:** We do not assume that strict equality is the optimal state.
>     *   **Empirical Evidence (Preliminary Study on Llama-3.2):** In fact, **during the early stages of our research**, we specifically investigated this hypothesis by conducting an experiment on **Llama-3.2** where we inserted a normalization layer **without training its parameters** (effectively acting as a fixed rescaling). The results showed that while **OCRBench reached 43.5** (showing gains in perception), performance on reasoning tasks stagnated (e.g., **ScienceQA at 58.29**, **MMStar at 32.08**).
>
>
>
> 3. **On Statistical Robustness and Scope:**
>     *   **Scope:** We have extended our validation to the **Qwen2.5-7B** architecture. The results confirm that the norm discrepancy persists in this state-of-the-art model and our method remains effective.
>
>
> 4.  **On Wording and Quantification:**
>     We apologize for the overstatement. We have revised the manuscript to replace "eliminated".
>
>
> **Response to Questions:**
>
>
> 1.  **On post-training norm disparity:**
>     Regarding the mitigation of norm disparity after training, we respectfully refer the reviewer to **Figure 4** in the manuscript. This figure explicitly visualizes the evolution of token norms, demonstrating that after applying our proposed LayerNorm, the norms of vision tokens are significantly reduced.
>
> 2.  **On missing standard deviations in Table 1:**
>     We apologize for the oversight regarding the mismatch between the caption and the table content. We have updated the tables  in the revised manuscript to include the **standard deviations** as originally intended.
>
> 3.  **On the increase of text embedding norms:**
>     We apologize for the lack of direct evidence in the main text. To substantiate the claim in lines 307–309, we have provided detailed experimental results in **Appendix D**.
>     *   **Findings:** The new results align perfectly with our theoretical analysis. Specifically, we observed that:
>         1.  In both **Qwen2-VL** and **Qwen2.5-VL**, the norms of the `<vision_start>` and `<vision_end>` tokens—which interact most intensively with image tokens—**increase significantly** after multimodal training.
>         2.  In **Qwen2.5-VL**, the mean norm of **all text tokens** shows a significant increase.
>
>
> 4.  **On the visualization of representational evolution:**
>     We appreciate this suggestion to analyze the temporal representational changes. We have conducted the additional analysis regarding the layer-wise similarity and time-evolution of representations. These specific results are now presented in **Appendix F** of the revised manuscript.

---

### Official Review · Reviewer_Wqxf · 2025-11-01

**Soundness:** 3
**Presentation:** 3
**Contribution:** 3
**Rating:** 8
**Confidence:** 2

**Summary:**

In this paper, the authors investigate a critical flaw in pre-normalized architectures within multimodal large language models (MLLMs): the significant disparity between high-norm visual tokens and low-norm textual tokens. This imbalance leads to a phenomenon referred to as “representational inertia,” where the semantics of high-norm visual tokens change at a considerably slower rate than those of text tokens, ultimately reducing the efficiency of cross-modal fusion between visual and textual features.

**Strengths:**

Identifies a Critical Issue: The paper highlights a significant and often overlooked issue regarding the norms of visual and textual tokens in multimodal large language models, shedding light on how this disparity can impair effective cross-modal integration.

Theoretical Contributions: By introducing the concept of "representational inertia," the authors provide a new theoretical framework for understanding the dynamics of feature representation in MLLMs. This contributes to the academic discourse on the challenges in multimodal learning.

Empirical Validation: The experiments conducted across multiple open-source multimodal models provide robust empirical support for the theoretical claims made, enhancing the credibility of the findings and their implications for model design.

Practical Solutions: The proposed solution of adding a layer normalization layer to enforce norm alignment is practical and relatively easy to implement, making it accessible for researchers and practitioners aiming to improve their multimodal models.

Enhanced Model Performance: The experimental results demonstrate significant improvements in model performance for both multimodal and text-only tasks, showcasing the effectiveness of the proposed intervention and its potential to advance the state of the art in multimodal understanding.

**Weaknesses:**

Lack of Broader Contextual Analysis: The paper primarily focuses on the norm differences without providing a comprehensive analysis of other factors that could affect multimodal performance, such as data quality or model architecture variations. This could limit a holistic understanding of the challenges in multimodal learning.

**Questions:**

While the proposed layer normalization solution effectively addresses norm differences between visual and textual tokens, how might this approach impact the model's ability to leverage subtle, nuanced variations within the visual or textual data that might be important for certain tasks, potentially leading to information loss or a homogenization of feature representations?

---

> ### Author Response · Authors · 2025-12-03
>
> **Response to Weakness:**
>
> We appreciate the reviewer's comment regarding the broader context of multimodal learning. However, we respectfully clarify that the primary objective of this paper is to specifically isolate and quantify the impact of **norm discrepancy** on model performance. To achieve this scientific rigor, we adopted a **controlled variable approach**. By deliberately keeping other factors—such as data quality and training strategies—constant, we were able to establish a clear causal link between norm discrepancy and optimization instability. While we agree that factors like data quality are significant, varying them in this study would have confounded our analysis of the norm effect.
>
> **Response to Question:**
>
> We thank the reviewer for raising the important concern regarding potential information loss. We believe our approach preserves, and even enhances, the model's ability to capture nuances for the following reasons:
>
> 1.  **Nature of the Discrepancy:** Intuitively, the extreme norm discrepancy observed is **not** an inherent feature representing semantic nuances between modalities. Instead, it is a **structural artifact** stemming from the **Pre-Norm architecture** . Correcting this does not discard information; it removes a structural bias that hinders optimization.
> 2.  **Empirical Evidence:** We validated our method across a wide range of **general and specialized benchmarks**. The consistent performance gains indicate that the model's ability to leverage subtle variations in data has been improved, not compromised.
> 3.  **Attention Analysis:** Our **attention visualization experiments** demonstrate that norm alignment leads to better-structured attention patterns. This indicates that the model is not homogenizing representations, but is instead able to attend to visual and textual tokens more effectively without one modality dominating the other due to sheer magnitude.

---

### Official Review · Reviewer_uNHA · 2025-11-01

**Soundness:** 3
**Presentation:** 2
**Contribution:** 2
**Rating:** 4
**Confidence:** 4

**Summary:**

This paper identifies a critical flaw in Pre-Norm MLLMs: a severe norm disparity between high-norm visual tokens and low-norm text tokens. The authors' theoretical analysis shows this imbalance creates an "asymmetric update dynamic". High-norm visual tokens exhibit "representational inertia", transforming semantically much slower than text tokens. This mismatch impairs cross-modal fusion and suppresses attention signals. The proposed solution is simple yet effective: inserting a single, carefully initialized LayerNorm after the visual projector to enforce norm alignment. This intervention yields significant performance gains on both multimodal benchmarks and, notably, on text-only evaluations like MMLU, suggesting a more holistically capable model.

**Strengths:**

1. The paper presents a novel perspective, using effective theoretical and empirical analysis (like inter-layer cosine similarity) to confirm the impact of "norm discrepancy" on the model's "asymmetric update dynamics".
2. The solution (inserting a single LayerNorm) is simple, effective, and easy to apply, achieving significant gains on multimodal and even text-only benchmarks.

**Weaknesses:**

The solution lacks validation on more model combinations. The paper's experiments are primarily on the SigLIP + Llama-3.2-3B combination. To prove generalizability, the authors should test on more structures, such as the LLaVA-1.5 (CLIP + Vicuna) combination.

**Questions:**

1. With sufficient training data, can the model learn to mitigate the "norm discrepancy" phenomenon on its own?
2. For models like Ovis[1] that use a "visual embedding table", does a similar "norm discrepancy" phenomenon exist?
3. In Table 2, the post-projector norms for KimiVL (4.78) and GLM-4.1V (4.58) are relatively close to the norm achieved by the author's method in Table 4 (2.2812). Does this imply these two models have already largely solved this problem with their more complex projectors?

[1] Ovis: Structural Embedding Alignment for Multimodal Large Language Model

---

> ### Author Response · Authors · 2025-12-03
>
> We thank the reviewer for raising these insightful points regarding the impact of data scaling and specific model architectures. We address your comments point-by-point below:
>
> 1.  **On whether increasing training data alleviates norm discrepancy (Qwen2.5-VL):**
>     We respectfully submit that scaling up training data does not necessarily resolve the norm discrepancy. To illustrate this, we cite **Qwen2.5-VL** as a counter-example. According to its technical report, the model was trained on a total of **4.1T multimodal tokens**. Despite this substantial training volume, the norm of image tokens remains high at **56.88**, whereas the text token norm is only **0.86**. This demonstrates that a significant norm discrepancy persists even under massive-scale training settings.
>
> 2.  **On the Ovis series:**
> We sincerely appreciate the reviewer for bringing the **Ovis series** to our attention. Following this suggestion, we have conducted experiments on the Ovis models and have updated **Section 4** of the revised manuscript accordingly. Our measurements reveal that a significant norm discrepancy persists within the Ovis architecture, where the average image embedding norm is approximately **64** compared to a text embedding norm of **1.38**. We attribute this substantial gap to the fact that the Ovis visual vocabulary is generated from the **SigLIP2 model**; given that the majority of parameters are preserved during this process [1], the model inherits the inherent high-magnitude characteristics of the original vision encoder.
> 3.  **On the Kimi and GLM series:**
>     We acknowledge that the norm discrepancies in the **Kimi and GLM series** are indeed mitigated. However, we hypothesize that this improvement is attributable to their use of more **complex adapter architectures or specific initialization strategies**, which facilitate better norm balancing even with fewer training tokens (e.g., Kimi utilized **2.3T tokens** according to their report, while GLM did not disclose this detail). Notably, their technical reports do not discuss the motivations or principles behind their adapter designs, nor do they specify the initialization methods used. **Therefore, we believe that this aspect remains a valuable direction for further exploration.** [2][3]
>
> [1] Ovis2.5 Technical Report
>
> [2] Kimi-VL Technical Report
>
> [3] GLM-4.5V and GLM-4.1V-Thinking: Towards Versatile Multimodal Reasoning with Scalable Reinforcement Learning

---

### Meta-Review · Area_Chair_TVUU · 2026-01-01

**Summary:**

This paper identifies and addresses an important yet overlooked failure mode in Pre-Norm MLLMs—a significant L2-norm mismatch between visual and text tokens, leading to asymmetric update dynamics and degraded cross-modal fusion. The work provides a clean theory-to-diagnostics-to-fix pipeline. The proposed intervention (a single, carefully initialized post-projector LayerNorm) is minimal, low-risk, and easy to adopt, with consistent empirical gains on multimodal benchmarks and the notable improvement on text-only evaluation (e.g., MMLU) suggesting broader benefits to model capability.

The main concerns are scope and robustness: initial experiments focused on a narrow set of model/encoder combinations with limited normalization/scale ablations, as well as simplified theoretical assumptions, alongside some presentation issues. The rebuttal and revision substantially addressed these points by adding new analyses/appendices, correcting table/caption issues, and extending validation/measurements to additional settings (e.g., Ovis and a Qwen2.5-based backbone), while clarifying why strict reproduction of proprietary large-scale training pipelines is infeasible.

Given the novelty of the diagnosis, the clarity of the mechanistic explanation, and the significant practical implications, I recommend acceptance with the expectation that future work will expand architectural coverage and deepen normalization/rescaling ablations.

**Reviewer Concerns:**

The main concerns are scope and robustness: initial experiments focused on a narrow set of model/encoder combinations with limited normalization/scale ablations, as well as simplified theoretical assumptions, alongside some presentation issues. The rebuttal and revision substantially addressed these points by adding new analyses/appendices, correcting table/caption issues, and extending validation/measurements to additional settings (e.g., Ovis and a Qwen2.5-based backbone), while clarifying why strict reproduction of proprietary large-scale training pipelines is infeasible.

**Reviewer Scores:**

Reviewer XYFm and Reviewer uNHA could raise the scores

---

### Decision · Program_Chairs · 2026-01-26

Accept (Poster)